# First detection and characterization of *mcr-1* colistin resistant *E. coli* from wild rat in Bangladesh

**Md. Wohab Ali**, **Susmita Karmakar, Kishor Sosmith Utsho, Ajran Kabir**, **Mohammad Arif**, **Md. Shafiqul Islam, Md. Tanvir Rahman, Jayedul Hassan***

Department of Microbiology and Hygiene, Bangladesh Agricultural University, Mymensingh, Bangladesh

* dr_jahid@bau.edu.bd

**Data Availability Statement:** All relevant data are within the manuscript and its Supporting information files. The genomic data could be accessed through the accession numbers provided in the manuscript.

## Abstract

Colistin resistance is a global concern warning for a one health approach to combat the challenge. Colistin resistant *E. coli* and their resistance determinants are widely distributed in the environment, and rats could be a potential source of these isolates and resistant determinants to a diverse environmental setting. This study was aimed to determine the presence of colistin resistant *E. coli* (CREC) in wild rats, their antimicrobial resistance (AMR) phenotypes, and genotypic analysis of *mcr-1* CREC through whole genome sequencing (WGS). A total of 39 rats were examined and CREC was isolated from their fecal pellets onto MacConkey agar containing colistin sulfate (1 μg/ mL). AMR of the CREC was determined by disc diffusion and broth microdilution was employed to determine MIC to colistin sulfate. CREC were screened for *mcr* genes (*mcr-1* to *mcr-8*) and phylogenetic grouping by PCR. Finally, WGS of one *mcr-1* CREC was performed to explore its genetic characteristics especially resistomes and virulence determinants. 43.59% of the rats carried CREC with one (2.56%) of them carrying CREC with *mcr-1* gene among the *mcr* genes examined. Examination of seventeen (17) isolates from the CREC positive rats (n = 17) revealed that majority of them belonging to the pathogenic phylogroup D (52.94%) and B2 (11.76%). 58.82% of the CREC were MDR on disc diffusion test. Shockingly, the *mcr-1* CREC showed phenotypic resistance to 16 antimicrobials of 8 different classes and carried the ARGs in its genome. The *mcr-1* gene was located on a 60 kb IncI2 plasmid. On the other hand, ARGs related to aminoglycosides, phenicols, sulfonamides, tetracyclines and trimethoprims were located on a 288 kb mega-plasmid separately. The *mcr-1* CREC carried 58 virulence genes including genes related to adhesion, colonization, biofilm formation, hemolysis and immune-evasion. The isolate belonged to ST224 and closely related to *E. coli* from different sources including UPEC clinical isolates from human based on cgMLST analysis. The current research indicates that rats might be a possible source of CREC, and the presence of *mcr-1* and other ARGs on plasmid increases the risk of ARGs spreading and endangering human health and other environmental components through this infamous pest.

**Funding:** This research was conducted with partial support from the research grants provided to JH, by Ministry of Education (MoE), Government of Bangladesh, Grant No. LS20191223; and Bangladesh Agricultural University Research Systems (BAURES), grant No. 2021/21/BAU. The funders did not play any role in the study design, data collection and analysis, decision to publish, or preparation of the manuscript.

**Competing interests:** The authors know no conflict of interest to declare.

## Introduction

Colistin is a reserve group of antibiotics. Colistin resistance became one of the key concerns due to its importance in treating MDR infections particularly with β-lactam and carbapenem resistance [1]. Colistin resistance is increasingly reported in the members of Enterobacteriaceae especially in *E. coli*. Although *E. coli* is a normal flora of the gastrointestinal tract of all warm-blooded animals, certain strain has become pathogenic through the acquisition of virulence genes or resistance through horizontal transfer and are associated with a range of intestinal and extra-intestinal infections [2, 3].

Colistin resistance is mediated by intrinsic adaptation, mutation or horizontal acquisition of colistin resistance [4]. Mobile colistin resistance (mediated by *mcr* genes) is a pivotal concern among the colistin resistance mechanisms due to rapid dissemination among different bacterial host through mobile genetic elements like plasmid and transposons [4, 5]. So far ten *mcr* genes (*mcr-1* to *mcr-10*) have been reported since the discovery of the first colistin resistant gene, *mcr-1* remained as the predominant one [6, 7]. Colistin resistant *E. coli* is widely distributed in the environment. *E. coli* carrying *mcr* genes was reported from different sources including human, street food, surface water, urban sludge, poultry meat and feces, and poultry farm environment [8–14]. However, no study has yet described the occurrence of colistin resistant especially *mcr-1 E. coli* from rodents such as rat in Bangladesh.

Rodents are reservoirs or carriers of MDR pathogens [15–19]. From one health perspective, rodents play important role in disseminating pathogens into the ecosystem to affect other hosts including wildlife, domestic animals and humans. Because of their inherent nature of dwelling human and animal food resources, rats are frequently seen in the household and farm environment especially poultry farms in Bangladesh. Recently, colistin resistant and MDR *E. coli* has been reported from fecal samples and poultry farm environments in Bangladesh and other part of the world [10, 14, 20] indicating a high chance of rats being exposed to such critical pathogens. And because of shared habitat, rats might be involved in the transmission of AMR including colistin resistant *E. coli* to human and livestock especially to poultry farms. Thus, the present study was conducted to examine the occurrence of colistin resistant *E. coli* (CREC), determination of *mcr* genotypes and characterization of a *mcr-1 E. coli* isolated from rats dwelling animal rearing facility through whole genome sequencing (WGS) based approach.

## Material and methods

### Ethical approval

This study design was approved by the Animal Welfare and Experimentation Ethics Committee, Bangladesh Agricultural University [Approval no. AWEEC/BAU/2021 (48)].

### Sample collection

A total of thirty-nine (39) wild rats were caught from the animal rearing facility (24.72544N90.43806E) during the period from October 2021 to June 2022. Fecal pellets were directly collected from the rat intestine after sacrificing by cervical dislocation. Three to four fecal pellets were collected from the distal part of the rat intestine in sterile Eppendorf tube and processed immediately for the isolation of CREC.

### Isolation of CREC

Fecal pellets were suspended in 1 mL Phosphate Buffer Saline (PBS) by repeated vortexing. Serial 10-fold dilution was prepared from the fecal suspension in PBS and 100 μL of the

suspension (non-diluted, 10-fold, 100-fold diluted) was spread onto MacConkey agar medium containing colistin sulfate 1 μg/mL (Fujifilm Wako Chemical Corporation, Osaka, Japan) [9]. After overnight (~16 h) incubation at 37˚C, *E. coli* specific lactose-fermenting colonies were streaked onto fresh MacConkey agar plates containing colistin sulfate (1 μg/mL) to obtain a pure culture. Genomic DNA was extracted from the *E. coli* suspected isolates by boiling method followed by PCR targeting the *malB* gene [21] using the primers furnished in S1 Table. Colistin resistance in the isolated *E. coli* was further confirmed by culturing the colonies in Luria Bertani (LB) broth containing colistin sulfate @ 2 μg/mL and stored at -20˚C for further uses.

## Detection and subtyping of mobile colistin resistance (*mcr*) genes

CREC isolated in this study was screened for *mcr* genes (*mcr-1* to *mcr-10*) by PCR. For *mcr-1* to *mcr-5* a multiplex PCR was employed [22]. The multiplex PCR was conducted in a final 20 μL reaction containing 10 μL one Taq® Quick-Load® 2X master mix (New England Biolabs Inc., Ipswich, MA, USA) and 5 pmol of each primer (S1 Table). *E. coli* strain 1ChBEc2mcr (*mcr-1* positive) [9] and *E. coli* strain ATCC25922 was used as the positive and negative controls, respectively. On the other hand uniplex PCR was performed for the detection of *mcr-6* to *mcr-10* following the in-house protocol developed by Awasthi et al. [Unpublished] and Göpel et al., 2023 [23].

## Phylogenetic grouping of *E. coli*

CREC isolated in this study were examined for their phylogenetic group by PCR targeting *chuA*, *yjaA*, and TspE4.C2 DNA fragments [24]. PCR was performed in a final 20 μL volume with One Taq® Quick-Load® 2X master mix (Biolabs) and 10 pmol of each primer followed by amplification at the conditions mentioned in S1 Table.

## Antibiotic susceptibility testing and minimum inhibitory concentration (MIC) determination

Disc diffusion test was performed to determine the antibiotic susceptibility of CREC isolated in this study [25]. A total of 21 antibiotics of 10 classes regularly prescribed in animal production and human clinical cases in Bangladesh, were tested (S2 Table). Each antibiotic was assessed at least three times to confirm the reproducibility of the results and *E. coli* strain ATCC25922 was used as the control strain. Isolates showing resistance to three or more classes of antibiotics were considered as MDR [26]. The MIC of colistin sulfate was determined by the broth microdilution method [25].

## Whole genome sequencing (WGS) and analysis

To explore the genomic characteristics and location of *mcr-1* gene, the *mcr-1* CREC strain isolated in this study (strain RJWEcMCR-1-BAU) was subjected to WGS on an Illumina Nextseq 550 platform (Illumina, CA, USA) commercially. The quality of FASTQ sequences and trimming of the low-quality reads were performed on Trimmomatic (Galaxy Version 0.38). Assembly of the trimmed reads were performed on a hybrid assembler Unicycler (Galaxy Version 0.4.8.0) followed by annotation through NCBI Prokaryotic Genomes Annotation Pipeline (PGAP). In addition, to explore the functional features RAST (Rapid Annotation using Subsystem Technology) server [27] was used.

The comparative ring image of the RJWEcMCR-1-BAU genome was prepared on BRIG (BRIG dist 0.95, [28]) and BacWGSTdb 2.0 server [29] was explored for the prediction of

sequence types, core genome MLST analysis (cgMLST) and source tracking of the genome. Detection of ARGs and VFGs were performed on the Comprehensive Antibiotic Resistance Database (CARD) [30], and VirulenceFinder [31] databases, respectively. The *E. coli* 1ChBEc2mcr genome was annotated against the PlasmidFinder [32] to detect plasmid replicons and the plasmid containing *mcr-1* gene was revealed through reference mapping on Geneious prime platform version 2022.1.1 (Biomatters). In addition, Proksee tools (Prokka, CARD Resistance Gene Identifier, Phigaro, mobileOG-db and CRISPR/Cas Finder [33] and Mobile Element Finder (MEF) version v1.0.3 (2020-10-09)) [34] were used to generate the circular images with distribution of contigs, CDS, ARGs and CRISPAR/Cas clusters, and mobile genetic elements (MGE) in the study genome. For mobile element detection 90% minimum coverage with 90% sequence identity and 30 maximum truncation filters were used.

## Results

### Occurrence of colistin resistant *mcr-1 E. coli* (*mcr-1* CREC) in rat

Selective culture in presence of colistin sulfate followed by PCR targeting the *malB* gene revealed 17 rat feces positive for CREC. PCR targeting *mcr* genes (*mcr-1* to *mcr-10*) revealed 1 rat carrying *mcr-1* and the others (16/17) carried non-mcr CREC according to the screening method. Overall, the occurrence of CREC and *mcr-1* CREC in the rats was recorded as 43.59% and 2.56%, respectively.

### Phylogenetic grouping of CREC

Most of the CREC isolated in this study belonged to pathogenic phylogroup of *E. coli* (64.70%) with 52.95% in the pathogenic phylogroup D and 11.76% in the group B2. On the other hand, 35.29% of the isolates belonged to commensal *E. coli* of phylogroup B1 (Table 1).

### Antibiotic susceptibility

Antibiotic susceptibility through disc diffusion identified 58.82% CREC as MDR (Table 2). Among these isolates, 100% were resistant to erythromycin, 52.94% to moxifloxacin, 41.18% were to ampicillin, amoxicillin and tetracycline. Surprisingly, the *mcr-1* CREC isolate RJWEcMCR-1-BAU was resistant to 16 antimicrobials of 8 different classes (Table 2). Broth microdilution revealed that all the non-*mcr* CREC isolates had an MIC of $\geq$ 2 μg/mL colistin sulfate, while the *mcr-1* CREC had an MIC of $\geq$ 8 μg/mL colistin sulfate.

### General characteristic of the RJWEcMCR-1-BAU genome

The genome was assembled to 126 contigs with an estimated size of 5203305 bp with 50.5% GC contents which corresponds to a standard *E. coli* genome. The longest and shortest contig size was 447963 and 268 bp, respectively with a mean length of 41296.1 bp (Table 3). The annotated genome consisted of 5039 coding sequences and 81 RNAs (rRNAs: 7, tRNAs: 65 and ncRNAs: 9). The genome comprises of eight (8) CRISPR regions and one Cas cluster with

**Table 1. Phylogenetic groups of colistin resistance *E. coli* isolated in this study.**

| Phylogenetic group | Number of isolates (n = 17) | Prevalence (%) |
|---|---|---|
| A | 0 | 0.0 |
| B1 | 6 | 35.29 |
| B2 | 2 | 11.76 |
| D | 9 | 52.94 |

**Table 2. Antimicrobial resistance pattern of the colistin resistance *E. coli* isolated in this study.**

| Number of antimicrobial classes | Resistance pattern | Number of isolates |
|---|---|---|
| 8 | AMX, AMP, AZM, CXM, CHL, CIP, DOX, ENR, ERY, LEV, MXF, NAL, NEO, STP, SXT, TET | 1 |
| 6 | AMX, AMP, AZM, DOX, ERY, MXF, STP, SXT, TET | 1 |
| 6 | AMX, AMP, DOX, ERY, MXF, STP, SXT, TET | 1 |
| 3 | AMX, AMP, CIP, ENR, ERY, LEV, MXF, NAL | 1 |
| 6 | AMX, AMP, ERY, MXF, STP, SXT, TET | 1 |
| 4 | AMX, AMP, AZM, ERY, MXF, TET | 2 |
| 2 | AMX, AMP, AZM, ERY | 1 |
| 4 | ERY, MXF, SXT, TET | 1 |
| 4 | ERY, MXF, NEO, TET | 1 |
| 3 | ERY, MXF, SXT | 1 |
| 2 | ERY, NEO | 1 |
| 2 | ERY, SXT | 1 |
| 1 | ERY | 4 |
| | Total | 17 |

AMP, Ampicillin; AMX, Amoxicillin; AZM, Azithromycin; CXM, Cefuroxime; CHL, Chloramphenicol; CIP, Ciprofloxacin; DOX, Doxycycline; ENR, Enrofloxacin; ERY, Erythromycin; LEV, Levofloxacin; MXF, Moxifloxacin; NAL, Nalidixic acid; NEO, Neomycin; STP, Streptomycin; SXT, Sulfamethoxazole-trimethoprim; TET, Tetracycline.

eight (8) gene signatures including *cas1, cas2, cas3, cas5, cas6, cas7, cse1, cse2* (S1 Fig) located in the contig no. 11. A total of seven (7) plasmids belonging to IncHI2, IncHI2A, IncI2(delta), IncN, IncX1 and p0111 replicon families with > 98% identity were detected in the genome by PlasmidFinder (S2 Fig). In addition, MEF identified 83 MGE in the genome with 10 insertion sequences of IS3, IS4, IS5, IS6 and ISL3 families (https://cge.food.dtu.dk//cgi-bin/webface.fcgi?jobid=6560055D0000568B5CCEA0E6, retrieved on 2023-11-24) complying filters with 90% coverage and 90% minimum identity and 30 maximum truncations. The genome also possesses four (4) prophage sequences, one belonging to Siphoviridae (47 kb) and three to

**Table 3. General features of the *E. coli* strain RJWEcMCR-1-BAU.**

| Genome features | |
|---|---|
| Sequence size | 5204818 |
| Number of contigs | 126 |
| GC content (%) | 50.5 |
| Shortest contig size | 268 bp |
| Mean sequence size | 41296.1 |
| Longest contig size | 447963 |
| N50 value | 193158 |
| L50 value | 11 |
| Number of subsystems | 381 |
| Number of coding sequences | 5039 |
| Number of RNAs | 81 (rRNAs—4 complete, 3 partial; tRNAs—65 and ncRNAs—9) |
| Sequence type (ST) | 224 |
| No. of ARGs | 76 |
| No. of virulence genes | 58 |

Myoviridae (17.6, 4.4 and 9.2 kb, respectively) (S1 Fig). The genome sequence is available in GenBank with an accession number of **JAUJUF000000000** (BioProject: **PRJNA992946** and BioSample: **SAMN36377858**). The version described in this paper is version **JAUJUF010000000**.

### Genetic relatedness of RJWEcMCR-1-BAU

The genome RJWEcMCR-1-BAU belongs to ST224 based on whole genome based cgMLST analysis and is closely related to *E. coli* reported from different parts of the world (Fig 1). The closest neighbor of this isolate is an *E. coli* (strain C103) reported from urine of a human in France which differ by 61 alleles. The isolate was also found very close to *E. coli* reported from chicken, pig, cow, dog and horses indicating its potential to interspecies dissemination. The RJWEcMCR-1-BAU genome was compared with *E. coli* strain Sakai (Accession no. GCA_000008865.2), the closest neighbor (*E. coli* strain C103, accession no. GCA_013047065.1), *E. coli* isolated from chicken meat in Bangladesh (strain 1ChBEc2mcr, accession no. GCA_027757765.1) and the sequences were found highly conserved; however, a significant number of non-homologous sequences were also evident throughout the genomes (Fig 2) which might be linked to the jumping or transposable sequences.

### Resistomes in the RJWEcMCR-1-BAU genome

The analysis of RJWEcMCR-1-BAU genome in CARD database revealed 76 antimicrobial resistance genes (ARGs) (Fig 3A) of 15 different classes with highest hits to fluoroquinolones followed by penems, cephalosporins and tetracyclines (Fig 3B). A number of genes responsible for various resistance mechanisms were detected in the genome including those involved in

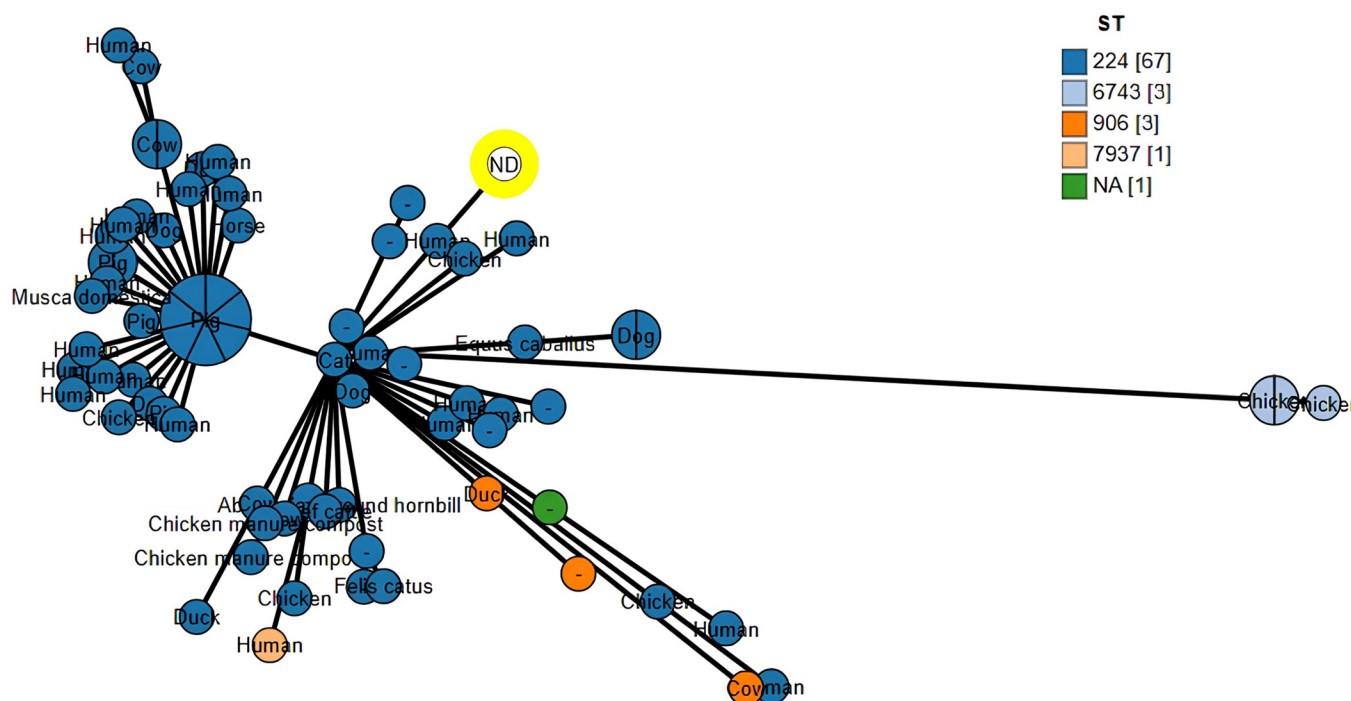

**Fig 1. Phylogenetic relationship between *E. coli* RJWEcMCR-1-BAU and close *E. coli* neighbors available in the NCBI GenBank database based on core genome multilocus sequence typing (cgMLST) analysis.** Node colors were assigned according to sequence type (ST). The yellow highlighted circle indicates the *mcr-1 E. coli* RJWEcMCR-1-BAU described in this study.

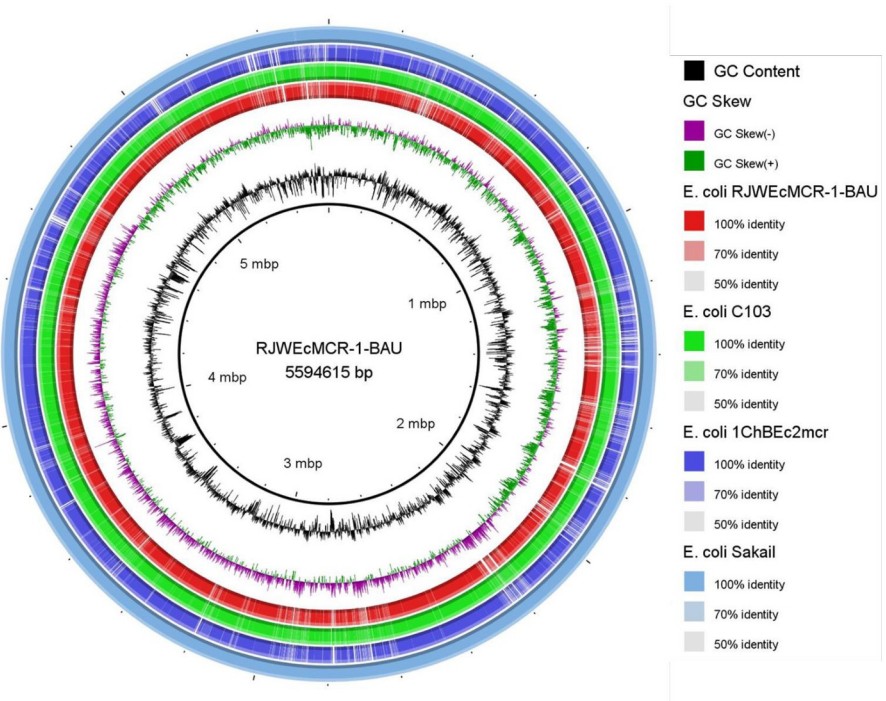

**Fig 2. Circular representation of the *E. coli* RJWEcMCR-1-BAU genome.** Circles (from inside to outside) 1 and 2 (GC content, black line; and GC skew, purple (- ve) and green (+ ve) lines); circle 3 (*E. coli* strain RJWEcMCR-1-BAU, red circle); circle 4 (*E. coli* strain C103, green circle, accession no. GCA_013047065.1); circle 5 (*E. coli* strain 1ChBEc2mcr, blue circle, accession no. GCA_027757765.1); circle 6 (*E. coli* strain Sakai; sky blue circle, accession no. GCA_000008865.2). BRIG 0.95 was used to build the circular representation and *E. coli* strain Sakai was used as the reference genome.

antibiotic efflux as the major one followed by antibiotic target alteration, inactivation, target replacement, target protection and reduced permeability (Fig 3C). The ARGs correspond to the phenotypic antibiotic resistance of this bacterium is shown in Fig 3 and Table 2. Colistin resistance gene *mcr-1* was extracted from the assembly, and the sequence was found identical to those reported from different sources and different plasmid types (S3 Fig).

## Virulence factors in the RJWEcMCR-1-BAU genome

RJWEcMCR-1-BAU carried 58 virulence related genes as identified by BacWGSTdb (47/58) and Virulence Finder (13/ 58) analysis. The bacterium carried genes associated with adhesion (*fdeC*), colonization and biofilm formation (*fimABCDFGHI*, *mrkABC*), enterobactin biosynthesis and iron acquisition (*entBCDEFS*, *fepABCDG*, *fes*), hemolysis (*hha*, *hlyE*), intracellular survival and immuno evasion (*ompA*, *rcsB*). Among the virulence factors adhesion, enterobactin production, hemolysins are previously described for their association with the virulence of uropathogenic *E. coli* (UPEC).

## Location of *mcr-1* and other plasmid mediated ARGs the RJWEcMCR-1-BAU genome

Raw reads of the RJWEcMCR-1-BAU genome were mapped with *mcr-1* gene carrying plasmids reported earlier (S3 Table). A plasmid sequence of 60,959 bp has been recovered from the sequences (Fig 4A, S1 Data) and named as pRJWEcMCR-1-BAU which is identical to an

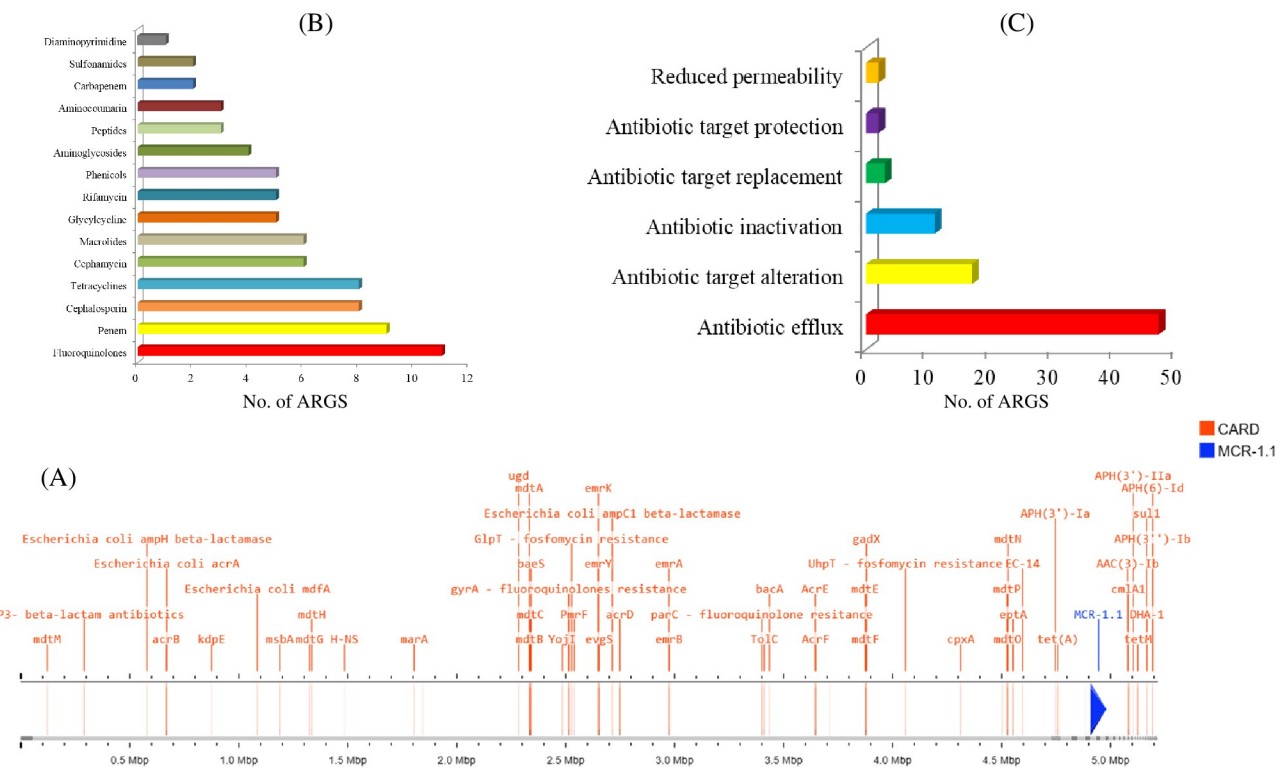

**Fig 3. An overview of the antimicrobial resistance genes (ARGs) (A), ARGs count against different antimicrobial classes (B), and resistance mechanisms (C), in *E. coli* RJWEcMCR-1-BAU.** The antimicrobial resistances genes (ARGs) and related pathways were explored through CARD (Comprehensive Antibiotic Resistance Database) are presented in red color. The MCR-1.1 gene is highlighted in blue color arrow head (A).

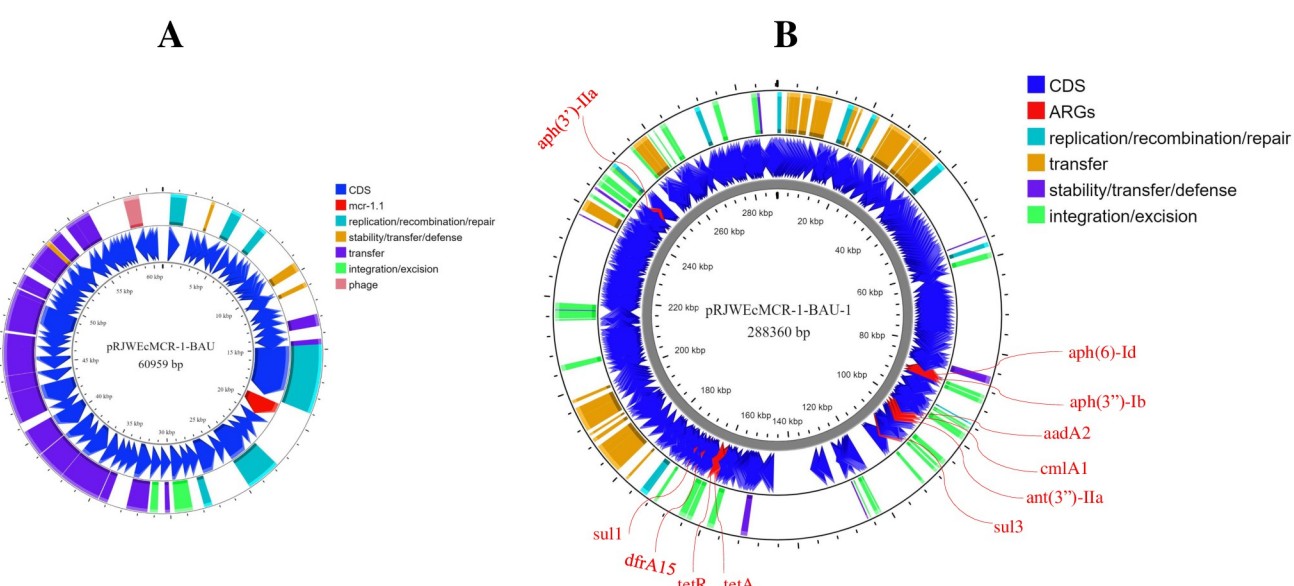

**Fig 4. Circular representation of the plasmid carrying *mcr-1.1* (A, pRJWEcMCR-1-BAU), and other plasmid related antimicrobial resistance genes (ARGs) (B, pRJWEcMCR-1-BAU-1) in RJWEcMCR-1-BAU.** The plasmid sequence was derived by mapping the raw reads with *E. coli* plasmid pHLJ109-25 (accession no. MN232198.1) and pEF7-18-51_3 (accession no. CP063490.1), respectively on Geneious Prime.

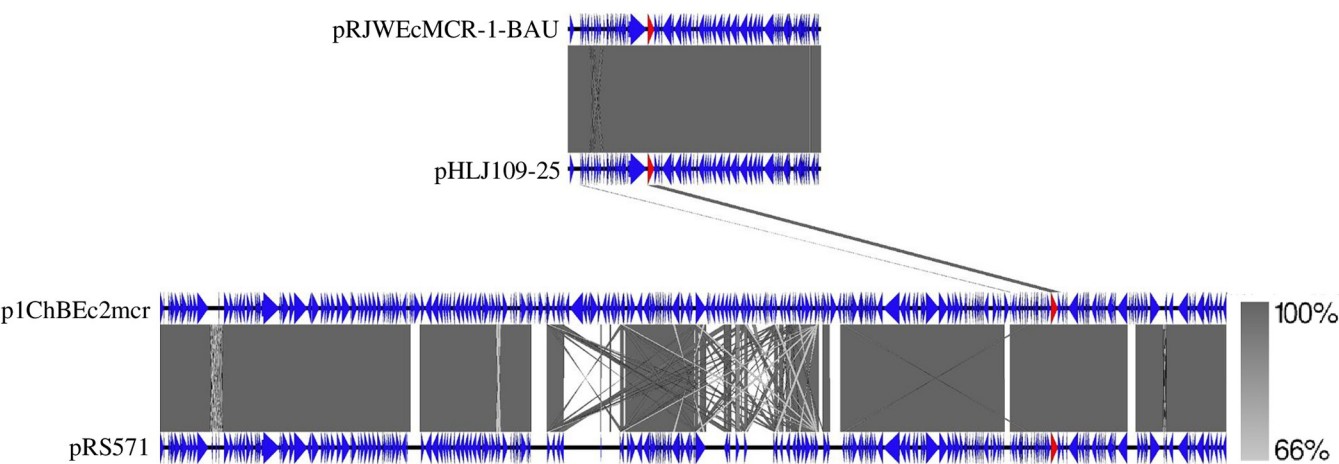

**Fig 5. Linear comparison of plasmid sequence RJWEcMCR-1-BAU.** The plasmid sequence was derived by mapping the raw reads with *E. coli* plasmid pHLJ109-25 (MN232198.1). Linear comparison was performed with *mcr-1.1* carrying plasmid reported from Bangladesh (pRS571-MCR-1.1, Accession no. CP034390; and p1ChBEc2mcr, Ali et al., 2023) on Easyfig version 2.2.5 (https://mjsull.github.io/Easyfig/files.html).

IncI2 (Delta) family plasmid pHLJ109-25 (accession no. MN232198.1) reported from chicken gut in China. The plasmid did not carry any other plasmid mediated ARGs. In addition, the plasmid sequence did not show homology to that reported from chicken meat and human clinical cases in Bangladesh except for the *mcr-1* gene indicating a newly reported plasmid carrying *mcr-1* genes (Fig 5). On the other hand, *aadA2, ant(3")-IIa, aph(3')-IIa, aph(3")-Ib, aph (6)-Id, cmlA1, drfA15, sul1, sul3, tetA* and *tetR* were located on a 288.36 kb plasmid pRJWEcMCR-1-BAU-1 belonging to the IncHI2A family as revealed by mapping against plasmid pSDC-F2_12BHI2 in *E. coli* (MH287085.1) (Fig 4B, S2 Data).

## Discussion

Rats are cosmopolitan and well adapted to diverse ecosystems. They are frequently encountered in human habitats and animal rearing facilities. Due to exposure to multiple environmental settings, they come in close contact with potential pathogens, and might become a potential source of pathogens and/ or resistant determinants especially AMR to humans and animals [17, 35, 36]. The occurrence of bacterial pathogens in rats caught from animal rearing facilities including poultry and pig farms has earlier been reported [37]. However, occurrences of bacterial pathogens, especially AMR in Rats remained untouched in Bangladesh. Recent reports on the occurrence of AMR in poultry farm environments [10, 14], urban sludges [11] and abundance of rats in the animal rearing facilities, prompted us to initiate this work. Colistin resistant *E. coli* are major public health concern. In this study we focused on the occurrence of colistin resistant *E. coli* in rat considering the increased trend of colistin resistance and potential of rats to transmit this notorious pathogen to humans or animals.

This study encountered colistin resistant *E. coli* in wild brown rats dwelling animal rearing facility and 64.7% of the isolates belonged to pathogenic phylogroup. Occurrence of colistin resistance in rat *E. coli* has been reported earlier [36] but to the best of our knowledge this is the first such report from Bangladesh. Remarkably, around 58.82% of the CREC are MDR and resistant to antimicrobials commonly used from human and animal practices in Bangladesh. Occurrence of MDR in CREC from rats is very alarming as they might act as the potential source of this resistant bacterium to their counterparts.

Colistin resistance is mediated by several mechanisms encoded by chromosomally mediated genes resembling to that of natural colistin resistant bacterium, and members of the *mcr*

gene family present on the horizontally transferable elements like plasmids and transposons [4, 5]. In this study, only one of the CREC carried *mcr-1* and the others were non-*mcr-1-10*. indicating that the non-*mcr-1-10* isolates might carry chromosomally mediated antimicrobial resistance as reported earlier from different part of the world including Bangladesh [10, 38]. To determine whether the *mcr-1* positive strain harbors chromosomal mutations crucial for colistin resistance, we scrutinized the amino acid sequences of PmrCAB, PhoPQ, PmrD, and MgrB, as outlined previously [39]. While no mutations were detected in PmrA, PmrC, PhoP, PhoQ, and MgrB, this strain exhibited mutations at D283G and Y358N in the PmrB, and K82T in the PmrD (S3 Data). Although these mutations have been identified in colistin-resistant *E. coli* strains, their experimental validation remains lacking [38, 40, 41]. However, the potential contribution of these mutations to the colistin resistance observed in our study strain cannot be disregarded. Hence, we recommend further studies to establish the association of these chromosomal mutation with the colistin resistance of *E. coli*. Interestingly, *mcr-1* carrying CREC had an elevated MIC ($\geq$ 8 µg/ mL) than the other CREC ($\geq$ 2 µg/ mL) which corresponds with the results from previous studies who described an elevated MIC in *mcr-1 E. coli* than non-mcr *E. coli* [10, 41]. Supporting the findings that production of *mcr-1* in *E. coli* leads to 4–8 fold increase in the MIC [42], the later CREC might belong to the non-mcr *E. coli*.

Mobile colistin resistance gene *mcr-1* has been reported on horizontally transferable plasmids as well on the chromosome flanked by a transposable element such as ISApI1 [4, 5, 9, 43]. In this study, the *mcr-1* gene was found to be located on a highly transferable IncI2 like plasmid as reported earlier [4]; however, no ISApI1 or similar transposon but *nikB* and *mobC* transfer related genes were located upstream of the *mcr-1* gene. Although the role of *nikB* and *mobC* in *mcr-1* gene transfer is not understood bacterial loss of ISApI1 have been published earlier [44] and speculated to be a part of bacterial adaptation to the environmental changes and essential for the maintenance of *mcr-1* gene [45]. Previously occurrence of *mcr-1* gene on IncF, IncHI2 plasmids was reported in Bangladesh [9, 46] but the occurrence of this in IncI2 plasmid is the first report incorporating additional information on the diversity of plasmids carrying *mcr-1* genes in this country.

In addition to resistance to colistin, bacterium RJWEcMCR-1-BAU showed resistance to 16 antimicrobials of 8 different classes. ARGs identification through CARD revealed the presence of all the ARGs related to the phenotypic resistance of this bacterium. Moreover, this bacterium carried ARGs conferring resistance to 15 different antimicrobial classes. Interestingly, blast homology search of the genome sequence indicated that most of the ARGs were associated with plasmid-like sequences. Although the location of all ARGs couldn't be confirmed due to short length sequencing, this study identified a 288 kb megaplasmid of highly transferable IncHI2 family that carries ARGs conferring resistance to aminoglycosides {*aadA2*, *ant (3")-IIa*, *aph(3')-IIa*, *aph(3")-Ib*, *aph(6)-Id*}, phenicols (*cmlA1*), sulfonamides (*sul1*, *sul3*), tetracyclines (*tetA*, *tetR*) and trimethoprims (*dfrA15*). Occurrence of MDR with *mcr-1 E. coli* has been reported earlier [9, 47], however, rats carrying these types of resistance determinants on plasmids are firstly reported in this study to the best of our knowledge. Occurrence of MDR *E. coli* carrying ARGs on horizontally transferable elements is worrisome and imposes a potential threat of transferring and disseminating these resistance determinants to human and animals through contaminated food, water and environment.

Source tracking through BacWGStdb and cgMLST revealed the isolate RJWEcMCR-1-BAU belonged to ST224 and closely related to *E. coli* isolated from humans and animals. Strikingly, the closest neighbor was a clinical strain of *E. coli* isolated from human urine from France (Accession no. NZ_JABBFG010000000) which differs from the study strain by only 61 alleles. More importantly, the study strain possesses virulence genes related to adhesion, colonization, biofilm formation, enterobactin biosynthesis and iron acquisition,

hemolysis and dodging host immunity which are important virulence determinants in ExPEC especially UPEC [48], indicating the strain RJWEcMCR-1-BAU might be a UPEC; however, their UPEC potential need to be further investigated in detailed through pheno-typic analysis.

The current investigation concentrated on the presence of CREC and the genomic charac-terization of a *mcr-1* CREC, isolated from a limited number of samples. Due to fund limitation, our primary goal was not to ascertain the prevalence of such CREC in rats; hence, we did not perform sample size calculations and screened a relatively small number of samples. Further-more, the rats involved in this research were obtained from a singular source, limiting the gen-eralizability of the study's outcomes. Consequently, we recommend further studies with a larger sample size, considering spatiotemporal variations, and encompassing a diverse array of rat species to bolster the robustness and applicability of our findings.

## Conclusion

Colistin-resistant *E. coli* (CREC) is major health concern. This is the first study in Bangladesh describing the occurrence of colistin-resistant *E. coli* (CREC) and the presence of *mcr-1* and other antimicrobial resistance genes on plasmids in CREC isolated from rats. The identifica-tion of MDR CREC in rats raises concerns about the potential dissemination of these patho-gens to various components of the one health system they interact with. However, further investigations with an expanded sample size, encompassing diverse locations and rat species from various habitats, are necessary to better understand and mitigate the threat of transfer-ring these pathogens and/or resistance determinants to different environmental components, including humans and animals.

## Supporting information

**S1 Fig. Circular view of the *E. coli* RJWEc-MCR-1-BAU genome.** A) Distribution of CDS, ARGs, CRISPR-Cas systems and prophages. The circular image shows the GC skew (Ring 1 from inside), GC content (Ring 2), CDS derived through Prokka annotation (Ring 3), ARGs from CARD analysis (Ring 4), CRISPR/Cas clusters (Ring 5) and prophages derived by Phi-garo tools (Ring 6). B) Linear view of the Cas-cluster in the RJWEc-MCR-1-BAU genome. Cir-cular view of the genome and other systems were prepared using Proksee tools (Prokka, CARD Resistance Gene Identifier, CRISPR/Cas Finder and Phigaro (https://proksee.ca/). (PPTX)

**S2 Fig. Plasmid replicon types detected in the *E. coli* RJWEcMCR-1-BAU genome through PlasmidFinder (http://cge.cbs.dtu.dk/services/PlasmidFinder/).** (PPTX)

**S3 Fig. Evolutionary relationship of *mcr-1* gene in strain RJWEcMCR-1-BAU.** The evolu-tionary history was inferred using the Neighbor-Joining method [1]. The optimal tree is shown. The tree is drawn to scale, with branch lengths in the same units as those of the evolu-tionary distances used to infer the phylogenetic tree. The evolutionary distances were com-puted using the p-distance method [2] and are in the units of the number of base differences per site. This analysis involved 35 nucleotide sequences. All ambiguous positions were removed for each sequence pair (pairwise deletion option). There were a total of 1674 positions in the final dataset. Evolutionary analyses were conducted in MEGA11 [3]. The strain described in this study is enclosed with red rectangle. (PPTX)

**S1 Table. Primers used in this study.**
(DOCX)

**S2 Table. Antimicrobial agents used in this study, along with their disc concentration, and classes.**
(DOCX)

**S3 Table. *E. coli* Plasmid sequences used as a reference in this study.**
(DOCX)

**S1 Data. pRJWEcMCR-1-BAU sequences.**
(TXT)

**S2 Data. pRJWEcMCR-1-BAU-1 sequences.**
(TXT)

**S3 Data. Analysis of chromosomal genes of RJWEcMCR-1-BAU for mutations conferring colistin resistance.**
(DOCX)

## Author Contributions

**Conceptualization:** Jayedul Hassan.

**Data curation:** Md. Wohab Ali, Ajran Kabir.

**Formal analysis:** Md. Wohab Ali, Susmita Karmakar, Kishor Sosmith Utsho, Ajran Kabir, Jayedul Hassan.

**Funding acquisition:** Jayedul Hassan.

**Investigation:** Md. Wohab Ali, Susmita Karmakar, Kishor Sosmith Utsho, Mohammad Arif.

**Methodology:** Md. Wohab Ali, Susmita Karmakar, Kishor Sosmith Utsho, Mohammad Arif, Jayedul Hassan.

**Project administration:** Jayedul Hassan.

**Resources:** Jayedul Hassan.

**Supervision:** Md. Shafiqul Islam, Md. Tanvir Rahman, Jayedul Hassan.

**Validation:** Md. Wohab Ali, Ajran Kabir, Md. Shafiqul Islam, Md. Tanvir Rahman, Jayedul Hassan.

**Visualization:** Md. Wohab Ali, Susmita Karmakar.

**Writing – original draft:** Md. Wohab Ali, Susmita Karmakar, Kishor Sosmith Utsho, Ajran Kabir, Mohammad Arif, Md. Shafiqul Islam, Md. Tanvir Rahman, Jayedul Hassan.

**Writing – review & editing:** Md. Tanvir Rahman, Jayedul Hassan.

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
