## [Decision Letter · Decision Letter 0]

28 Dec 2023

PONE-D-23-39379

First detection and characterization of mcr-1 colistin resistant E. coli from wild rat in Bangladesh

PLOS ONE

Dear Dr. Hassan,

Thank you for submitting your manuscript to PLOS ONE. After careful consideration, we have decided that your manuscript does not meet our criteria for publication and must therefore be rejected.

The reviewer has aptly pinpointed several relevant gaps in the study. Moreover, it is important to note an additional limitation concerning the study's scope. Specifically, the study is constrained by a relatively limited number of participants and isolates analyzed through Whole Genome Sequencing (WGS).

Taking these limitations into account collectively, I am sorry that my assessment of the current manuscript cannot be more positive, but hope that you appreciate the reasons for this decision.

Kind regards,

Marwan Osman

Academic Editor

PLOS ONE

Reviewers' comments:

Reviewer's Responses to Questions

**Comments to the Author**

1. Is the manuscript technically sound, and do the data support the conclusions?

Reviewer #1: No

2. Has the statistical analysis been performed appropriately and rigorously? 

Reviewer #1: N/A

3. Have the authors made all data underlying the findings in their manuscript fully available?

Reviewer #1: No

4. Is the manuscript presented in an intelligible fashion and written in standard English?

Reviewer #1: Yes

5. Review Comments to the Author

Reviewer #1: In this manuscript, Ali et al. reported the first detection and characterization of mcr-1 colistin-resistant E. coli in wild rats in Bangladesh. The prevalence of colistin-resistant E. coli has been extensively documented across various sources, including humans, animals, and the environment in Bangladesh. Given its widespread presence in the environment, it is not unexpected for rats, as environmental scavenger, to harbor these organisms. Consequently, the first report of mcr-1-positive colistin-resistant E. coli in Bangladesh does not significantly contribute to the existing knowledge in this domain. A more compelling approach would involve systematic surveillance, accounting for spatiotemporal variations in sampling and considering the diversity of rat species commonly found in household premises in Bangladesh.

Additionally, the research is constrained by a limited sample size, with only 39 rats subjected to testing. Furthermore, all the rats included in the study were captured from a single location, thereby restricting the generalizability of the study findings.

The methodology used for isolating colistin-resistant E. coli raises concerns. The concentration of colistin sulfate utilized on the MacConkey plate falls significantly below the MIC of colistin for E. coli. The rationale behind selecting this concentration (1 µg/ml) remains unclear. Was this choice informed by prior studies or established through in-house experiments? Selection of isolates in culture plates with a lower concentration of colistin may facilitate the growth of colistin-sensitive or intermediately resistant isolates apart from the colistin-resistant isolates. Therefore, it is imperative to confirm resistance through determining MIC. In this study, the authors conducted MIC assessments for all isolates reported to be colistin resistant obtained from the primary plate with lower concentration of colistin sulfate. However, with the exception of one isolate, which also tested positive for the mcr-1 gene, the MIC for all other isolates was ≤2 µg/ml, indicating an intermediate category per CLSI guidelines. Furthermore, the study did not specify how many of these isolates had an MIC of <2 µg/ml. Consequently, the reported prevalence of 43.59% for colistin-resistant E. coli in the study appears to be an overestimation.

The discussion section should include a description of the study's limitations.

6. PLOS authors have the option to publish the peer review history of their article (what does this mean?). If published, this will include your full peer review and any attached files.

Reviewer #1: No

- - - - -

---

## [Author Response · Author response to Decision Letter 0]

11 Jan 2024

Response to Reviewers

The authors express sincere gratitude to the reviewer for his insightful perspective and valuable suggestions on the manuscript. We have thoroughly addressed all queries and incorporated suggested modifications. Below, we have provided a point-by-point response to the reviewer's comments for your thoughtful consideration-

1. Is the manuscript technically sound, and does the data support the conclusions?

Reviewer #1: No.

Response: We appreciate the input from the reviewer and acknowledge that they may hold a unique opinion and interpretation. Despite providing a comprehensive conclusion based on our research findings, we have made modifications to the conclusion for further consideration.

2. Has the statistical analysis been performed appropriately and rigorously?

Reviewer #1: N/A

3. Have the authors made all data underlying the findings in their manuscript fully available?

Reviewer #1: NO. 

Response: To the best of our knowledge, we have included all pertinent data and corresponding links that are publicly accessible for this manuscript, as indicated by the GenBank accession numbers provided in the document. We are uncertain about the specific data the reviewer is referring to as being unavailable, as we have taken extensive measures to ensure comprehensive access. Furthermore, we are readily available to furnish any specific information that the reviewer deems to be missing in the submitted manuscript.

4. Is the manuscript presented in an intelligible fashion and written in standard English?

Reviewer #1: Yes. Response: Thanks for his positive comments on our English writing standard.

5. Review Comments to the Author

Reviewer #1: In this manuscript, Ali et al. reported the first detection and characterization of mcr-1 colistin-resistant E. coli in wild rats in Bangladesh. The prevalence of colistin-resistant E. coli has been extensively documented across various sources, including humans, animals, and the environment in Bangladesh. Given its widespread presence in the environment, it is not unexpected for rats, as environmental scavengers, to harbor these organisms. Consequently, the first report of mcr-1-positive colistin-resistant E. coli in Bangladesh does not significantly contribute to the existing knowledge in this domain. A more compelling approach would involve systematic surveillance, accounting for spatiotemporal variations in sampling and considering the diversity of rat species commonly found in household premises in Bangladesh.

Response: 

We acknowledge the reviewer's observation; however, our primary focus centers on investigating the presence of colistin-resistant E. coli (CREC), particularly mcr-CREC, in rats. This is crucial given their role as potential mediators in the spread of pathogens to humans and animals. While it is conceivable for rats to acquire such resistant bacteria from their environment, we question whether theoretical knowledge alone should guide inferences and decisions without practical validation.

To our knowledge, there have been no reports of colistin-resistant E. coli from wild rats in Bangladesh or neighboring countries. Furthermore, our findings highlight the presence of plasmids carrying mcr-1 and other antimicrobial resistance genes, representing the first report of its kind in this country. The occurrence of such a resistant pathogen in rats holds significant importance, considering the adaptability of these scavengers to human and animal environments. Rats can potentially serve as a source of this pathogen to their counterparts.

We believe that these data are immensely important in elucidating the genetic features of mcr-1 CREC, tracing potential sources, and paving the way for further studies to identify potential mediators and formulate effective control strategies.

While we acknowledge that a more systematic surveillance, along with an exploration of spatiotemporal variations and diversity among rat species, could offer valuable insights, it's crucial to recognize our constraints in terms of facilities and funding. It's essential to note that even studies published in reputable journals like Nature or Science have inherent limitations. We are actively planning to secure additional funding for further exploration in the future. The acceptance of our manuscript would not only facilitate smoother access to funds but also pave the way for more in-depth research in this area.

Reviewer #1: Additionally, the research is constrained by a limited sample size, with only 39 rats subjected to testing. Furthermore, all the rats included in the study were captured from a single location, thereby restricting the generalizability of the study findings.

Response:

We concur with the reviewer's observation regarding the small sample size and the limitation of being collected from a single location, which indeed limits the generalizability of our study findings. Despite these acknowledged limitations, we emphasize the significant implications this study holds for guiding future research endeavors and securing funding for more extensive and thorough investigations. Furthermore, we have explicitly outlined these limitations and outlined potential future directions in our discussion.

Reviewer #1: The methodology used for isolating colistin-resistant E. coli raises concerns. The concentration of colistin sulfate utilized on the MacConkey plate falls significantly below the MIC of colistin for E. coli. The rationale behind selecting this concentration (1 µg/ml) remains unclear. Was this choice informed by prior studies or established through in-house experiments? Selection of isolates in culture plates with a lower concentration of colistin may facilitate the growth of colistin-sensitive or intermediately resistant isolates apart from the colistin-resistant isolates. Therefore, it is imperative to confirm resistance through determining MIC. In this study, the authors conducted MIC assessments for all isolates reported to be colistin resistant obtained from the primary plate with lower concentration of colistin sulfate. However, with the exception of one isolate, which also tested positive for the mcr-1 gene, the MIC for all other isolates was ≤2 µg/ml, indicating an intermediate category per CLSI guidelines. Furthermore, the study did not specify how many of these isolates had an MIC of <2 µg/ml. Consequently, the reported prevalence of 43.59% for colistin-resistant E. coli in the study appears to be an overestimation.

Response: 

Thank you for addressing this concern. As per CLSI guidelines, polymyxins resistance is categorized as intermediate and resistant, with no susceptible category. It's crucial to note that the intermediate category does not imply sensitivity but rather resistance to the antimicrobial. Our utilization of 1 µg/ml colistin sulfate in plates for CREC isolation is not a novel approach and has been documented previously (doi: 10.1016/j.ijfoodmicro.2022.110065).

In our study, we directly inoculated samples onto plates without enrichment, resulting in minimal colony growth—sometimes only 2 to 3 colonies from an undiluted suspension. This suggests a low likelihood of cultivating non-colistin resistant E. coli in this condition. We found that using 2 µg/ml or higher colistin concentrations on plates decreases the likelihood of identifying intermediate-sensitive/resistant CREC. To confirm resistance, we examined all colonies on a plate for growth in LB broth containing 2 µg/ml colistin sulfate. All colonies obtained on MacConkey agar plates were confirmed as resistant to ≥2 µg/ml. Contrary to a typo indicating ≤2 µg/ml in the manuscript, all CREC isolates, except the mcr-1 CREC, had an MIC of ≥2 µg/ml colistin sulfate. We appreciate the reviewer's keen observation, and the typo has been corrected.

In this manuscript, we reported the occurrence of CREC rather than prevalence due to the limited number of samples, making prevalence estimation challenging. Occurrence was calculated based on the isolation of CREC with MIC ≥2 µg/ml. Although not surprising, a more extensive sample size might yield different results, warranting further studies. Unfortunately, due to funding constraints, we couldn't conduct a more systematic surveillance, considering spatiotemporal variations and diversity among rat species. We acknowledge these limitations in the updated manuscript and plan for a more detailed study when funds become available. Your understanding of these limitations is greatly appreciated.

---

## [Decision Letter · Decision Letter 1]

16 Feb 2024

PONE-D-23-39379R1First detection and characterization of mcr-1 colistin resistant E. coli from wild rat in BangladeshPLOS ONE

Dear Dr. Hassan,

Thank you for submitting your manuscript to PLOS ONE. After careful consideration, we feel that it has merit but does not fully meet PLOS ONE’s publication criteria as it currently stands. Therefore, we invite you to submit a revised version of the manuscript that addresses the points raised during the review process.

We look forward to receiving your revised manuscript.

Kind regards,

Zhi Ruan, Ph.D.

Academic Editor

PLOS ONE

Journal Requirements:

When submitting your revision, we need you to 
address these

additional requirements.

1. Please ensure that your manuscript meets PLOS ONE's style

requirements, including those for file naming. The PLOS ONE style templates can

be found at

and

2. Did you know that

depositing data in a repository is associated with up to a 25% citation

advantage (https://doi.org/10.1371/journal.pone.0230416)? If you’ve not already

done so, consider depositing your raw data in a repository to ensure your work

is read, appreciated and cited by the largest possible audience. You’ll also

earn an Accessible Data icon on your published paper if you deposit your data

in any participating repository (https://plos.org/open-science/open-data/#accessible-data).

3. We suggest you thoroughly copyedit your manuscript for

language usage, spelling, and grammar. If you do not know anyone who can help

you do this, you may wish to consider employing a professional scientific

editing service.

Whilst you may use

any professional scientific editing service of your choice, PLOS has partnered

with both American Journal Experts (AJE) and Editage to provide discounted

services to PLOS authors. Both organizations have experience helping authors

meet PLOS guidelines and can provide language editing, translation, manuscript

formatting, and figure formatting to ensure your manuscript meets our

submission guidelines. To take advantage of our partnership with AJE, visit the

AJE website (http://aje.com/go/plos) for a 15% discount off AJE services. To

take advantage of our partnership with Editage, visit the Editage website

(www.editage.com) and enter referral code PLOSEDIT for a 15% discount off

Editage services. If the PLOS editorial team finds any language issues in text

that either AJE or Editage has edited, the service provider will re-edit the

text for free.

Upon resubmission,

please provide the following:

The name of the

colleague or the details of the professional service that edited your manuscript

A copy of your manuscript showing your changes

by either highlighting them or using track changes (uploaded as a *supporting

information* file)

A clean copy of the

edited manuscript (uploaded as the new manuscript file)”"

“This research was

conducted with partial support from the research grants provided to JH, by

Ministry of Education (MoE), Government of Bangladesh, Grant No. LS20191223;

and Bangladesh Agricultural University Research Systems (BAURES), grant No.

2021/21/BAU. The funders did not play any role in the study design, data

collection and analysis, decision to publish, or preparation of the manuscript.”

Please provide an amended statement that declares *all* the

funding or sources of support (whether external or internal to your

organization) received during this study, as detailed online in our guide for

authors at http://journals.plos.org/plosone/s/submit-now. Please also include the statement “There was

no additional external funding received for this study.” in your updated

Funding Statement.

Please include your amended Funding Statement within your

cover letter. We will change the online submission form on your behalf.

5. Thank you for stating the following in the

Acknowledgments Section of your manuscript:

“The authors are indebted to the Ministry of Education (MoE),

Government of Bangladesh (Project No. LS20191223), and Bangladesh Agricultural

University Research Systems (BAURES) (Project No. 2021/21/BAU) for providing partial

fund for this research.”

We note that you

have provided additional information within the Acknowledgements Section that

is not currently declared in your Funding Statement. Please note that funding

information should not appear in the Acknowledgments section or other areas of

your manuscript. We will only publish funding information present in the

Funding Statement section of the online submission form.

Please remove any funding-related text from the manuscript

and let us know how you would like to update your Funding Statement. Currently,

your Funding Statement reads as follows:

“This research was conducted with partial support from the

research grants provided to JH, by Ministry of Education (MoE), Government of

Bangladesh, Grant No. LS20191223; and Bangladesh Agricultural University

Research Systems (BAURES), grant No. 2021/21/BAU. The funders did not play any

role in the study design, data collection and analysis, decision to publish, or

preparation of the manuscript.”

Please include your amended statements within your cover

letter; we will change the online submission form on your behalf.

6. We note that you have included the phrase “data not

shown” in your manuscript. Unfortunately, this does not meet our data sharing

requirements. PLOS does not permit references to inaccessible data. We require

that authors provide all relevant data within the paper, Supporting Information

files, or in an acceptable, public repository. Please add a citation to support

this phrase or upload the data that corresponds with these findings to a stable

repository (such as Figshare or Dryad) and provide and URLs, DOIs, or accession

numbers that may be used to access these data. Or, if the data are not a core

part of the research being presented in your study, we ask that you remove the

phrase that refers to these data.

Additional Editor Comments (if provided):

Please carefully address the reviewers' comments from the two reviewers. Also, please cite the bioinformatics tools used in your study. For example, Line 129, BacWGSTdb 2.0 server.

Reviewers' comments:

Reviewer's Responses to Questions

**Comments to the Author**

1. If the authors have adequately addressed your comments raised in a previous round of review and you feel that this manuscript is now acceptable for publication, you may indicate that here to bypass the “Comments to the Author” section, enter your conflict of interest statement in the “Confidential to Editor” section, and submit your "Accept" recommendation.

Reviewer #2: (No Response)

Reviewer #3: (No Response)

2. Is the manuscript technically sound, and do the data support the conclusions?

Reviewer #2: Yes

Reviewer #3: Yes

3. Has the statistical analysis been performed appropriately and rigorously? 

Reviewer #2: N/A

Reviewer #3: N/A

4. Have the authors made all data underlying the findings in their manuscript fully available?

Reviewer #2: No

Reviewer #3: No

5. Is the manuscript presented in an intelligible fashion and written in standard English?

Reviewer #2: Yes

Reviewer #3: No

6. Review Comments to the Author

Reviewer #2: This manuscript has been revised on first round. I have some comments to the authors below:

1. Why the author detect only mcr-1 to mcr-8? Why not mcr-9 and mcr-10?

2. Although the author detected one E.coli isolate which carrying mcr-1 and colistin was also resistance, but it is not guarantee that this phenomenon of colistin resistance was due to mcr-1.

Chromosomal-mediated colistin resistant genes are also important. I recommend the author should extensive additional analysis of these chromosomal-gene mutations because of you have WGS data of mcr-1-harboring isolate. You can follow this paper as a guideline for these genes.

3. Please discuss your result of (2) in discussion part.

Binsker U, Käsbohrer A, Hammerl JA. Global colistin use: a review of the emergence of resistant Enterobacterales and the impact on their genetic basis. FEMS Microbiol Rev. 2022;46(1):fuab049. doi:10.1093/femsre/fuab049

Reviewer #3: lines 90. 91 - Is MacConkey selective for E. coli?

line 168-169 - A total of seven (7) plasmids belonging to IncHI2, IncHI2A, IncI2(delta), IncN, IncX1 169 and p0111 replicon families with > 98% identity were detected in the genome by PlasmidFinder. -Does the software detect plasmid replicons or plasmids?

158 - Antibiotic susceptibility through disc diffusion identified 58.82 % CREC as MDR (Table 2). - Table 2 shows antibiotic resistance patterns with no mention of MDR statistics.

Fig 1: Has bad resolution

Fig 3: An overview of the antimicrobial resistance genes (ARGs) (A), resistant antibiotics (B), and resistance mechanisms (C), in E. coli RJWEcMCR-1-BAU. - What do you mean by resistant antibiotics?

Fig 4: The plasmid sequence was derived by mapping the raw reads with E. coli plasmid pHLJ109-25 - This is not clear. Where the authors able to map a plasmid using illumina sequencing. Because of the sequencing depth it is generally impossible to map entire plasmids using the technique. Hence, there is reference to plasmid replicons as these can be predicted from the contigs. The authors seem to have been able to accurately identify two plasmids one carrying the mcr- 1 gene (The mcr-1 gene was located on a 60 kb IncI2 plasmid.) and another one carrying a myriad of ARGs (on a 288 kb mega-plasmid separately). Are the authors reporting similarity of their raw reads with plasmids or did they actually map plasmids as reported?

There's need for clarity on how many E. coli isolates were obtained from the 39 rats. How many isolates from each rat. Clarity is also required on the the 17 CREC. The MIC results of all 17 should be presented. The mcr-1 positive isolate was clearly colistin resistant (MIC of ≥ 8 µg/ml) but the others were susceptible ≤ 2 µg/mL according to EUCAST guidelines. Why were all 17 isolates considered CREC?

7. PLOS authors have the option to publish the peer review history of their article (what does this mean?). If published, this will include your full peer review and any attached files.

Reviewer #2: No

Reviewer #3: **Yes: **Joshua Mbanga

---

## [Author Response · Author response to Decision Letter 1]

28 Mar 2024

Response to Reviewers

The authors sincerely appreciate the reviewer for providing insightful perspectives and valuable suggestions on the manuscript. We have carefully addressed all queries and incorporated the suggested modifications. Below, we present a point-by-point response to the reviewer's comments for your thoughtful consideration:

Reviewer #2: This manuscript has been revised on first round. I have some comments to the authors below:

1. Why the author detect only mcr-1 to mcr-8? Why not mcr-9 and mcr-10?

Response: Thank you for your nice comment. The prevalence of mcr-1 is extensive, whereas occurrences of mcr-2 to mcr-10 have been reported sporadically. Given the laboratory resources available during the experiment, we initially investigated the presence of mcr-1 to mcr-8. However, in response to the reviewer's query and to prevent any confusion, we conducted additional analysis on the samples for mcr-9 and mcr-10. Unfortunately, none of the isolates tested positive for the mcr-9 or mcr-10 genes. We have diligently updated this information in the manuscript to present a comprehensive overview (Ln. 96, 102-103, 146, 248-249 in the Track changed version, and Supplementary Table 1). 

2. Although the author detected one E. coli isolate which carrying mcr-1 and colistin was also resistance, but it is not guarantee that this phenomenon of colistin resistance was due to mcr-1.

Chromosomal-mediated colistin resistant genes are also important. I recommend the author should extensive additional analysis of these chromosomal-gene mutations because of you have WGS data of mcr-1-harboring isolate. You can follow this paper as a guideline for these genes.

3. Please discuss your result of (2) in discussion part.

Binsker U, Käsbohrer A, Hammerl JA. Global colistin use: a review of the emergence of resistant Enterobacterales and the impact on their genetic basis. FEMS Microbiol Rev. 2022;46(1):fuab049. doi:10.1093/femsre/fuab049

Response: We extend our sincere gratitude to the reviewer for their valuable suggestion. Our investigation delved into the whole-genome sequencing (WGS) and subsequent analysis of chromosomal genes potentially implicated in the development of colistin resistance. Following the guidance provided in the manuscript mentioned by the reviewer, we meticulously scrutinized the aa sequences of PmrCAB, PmrD, PhoPQ, and MgrB. 

Our analysis revealed no mutations in the aa sequences of PmrA, PmrC, PhoP, PhoQ, and MgrB. However, we did detect two mutations in PmrB at positions D283G and Y358N, along with one mutation in PmrD at position K82T. These mutations were previously identified in colistin-resistant strains of E. coli, albeit lacking experimental confirmation.

It's worth noting that we did not conduct experimental validation in this study, thus we cannot conclusively rule out the potential chromosomal contribution to the colistin resistance observed in our strain. We have elaborated on this matter in our manuscript (Ln. 250-260, Track changed version) and provided the gene sequences as supplementary data (Supplementary data 3) for your thoughtful consideration.

Reviewer #3: lines 90. 91 - Is MacConkey selective for E. coli?

Response: MacConkey agar is indeed selective for Gram-negative bacteria, particularly for members of the Enterobacteriaceae family, such as Escherichia coli. One reason why MacConkey agar is commonly used for isolating E. coli from crude samples like feces or food is because it not only selects for Gram-negative bacteria but also differentiates them based on their ability to ferment lactose. E. coli, being a lactose fermenter, produces colonies that appear pink to red on MacConkey agar. This selective and differential medium helps in the isolation and identification of E. coli from mixed microbial populations. Other than MacConkey, EMB agar (Eosin Methylene Blue agar) is another selective and differential medium commonly used for isolating Gram-negative bacteria, including E. coli. However, both media are suitable for isolating E. coli. MacConkey agar is commonly used in our laboratory and long been used in our lab for its simplicity and effectiveness in differentiating lactose fermenters, which is particularly useful when isolating E. coli from samples with potentially high microbial diversity, such as feces or certain food samples. 

line 168-169 - A total of seven (7) plasmids belonging to IncHI2, IncHI2A, IncI2(delta), IncN, IncX1 169 and p0111 replicon families with > 98% identity were detected in the genome by PlasmidFinder. -Does the software detect plasmid replicons or plasmids?

Response: PlasmidFinder does not directly detect entire plasmids, it provides information about the presence of specific plasmid replicons within bacterial isolates, which can be indicative of the presence of certain types of plasmids. Replicon sequences from 559 fully sequenced plasmids associated with the family Enterobacteriaceae in the NCBI nucleotide database were collected to build a consensus database for integration into PlasmidFinder that can be used for replicon sequence analysis of raw, contig group, or completely assembled and closed plasmid sequencing data. The PlasmidFinder database currently consists of 116 replicon sequences that match with at least at 80% nucleotide identity all replicon sequences identified in the 559 fully sequenced plasmids.

158 - Antibiotic susceptibility through disc diffusion identified 58.82 % CREC as MDR (Table 2). - Table 2 shows antibiotic resistance patterns with no mention of MDR statistics.

Response: We appreciate your gracious feedback. We would like to draw your attention to Table 2, specifically the left panel, where we have highlighted the Number of antimicrobial classes. As described in the manuscript, specifically lines 115 to 116 (Track changed version), we defined isolates exhibiting resistance to three or more classes of antibiotics as MDR. We trust this clarification fulfills the necessary criteria.

Fig 1: Has bad resolution

Response: Figure 1 has been optimized to comply with the journal's specifications. Following your suggestion, we have further enhanced the resolution of Figure 1.

Fig 3: An overview of the antimicrobial resistance genes (ARGs) (A), resistant antibiotics (B), and resistance mechanisms (C), in E. coli RJWEcMCR-1-BAU. - What do you mean by resistant antibiotics?

Response: Apologies for any confusion caused by our previous heading. We've revised it accordingly to clarify that "resistant antibiotic" refers to antimicrobial resistance genes (ARGs) detected across various antimicrobial classes. Thank you for your understanding. 

Fig 4: The plasmid sequence was derived by mapping the raw reads with E. coli plasmid pHLJ109-25 - This is not clear. Where the authors able to map a plasmid using illumina sequencing. Because of the sequencing depth it is generally impossible to map entire plasmids using the technique. Hence, there is reference to plasmid replicons as these can be predicted from the contigs. The authors seem to have been able to accurately identify two plasmids one carrying the mcr- 1 gene (The mcr-1 gene was located on a 60 kb IncI2 plasmid.) and another one carrying a myriad of ARGs (on a 288 kb mega-plasmid separately). Are the authors reporting similarity of their raw reads with plasmids or did they actually map plasmids as reported?

Response: We express our sincere gratitude to the reviewer for their insightful query. The advancements in technology over recent years have been remarkable, particularly in the realm of genomics. Mapping high-throughput data has become a swift process with modern genomic workbenches such as Biomatters (Geneious Prime) and CLC Genomics Workbenches.

In this study, we initially conducted a BLAT search to identify sequence similarity of the mcr-1.1 gene and other antimicrobial resistance-containing contigs. Plasmids exhibiting 100% similarity were then downloaded, and the raw reads were subsequently mapped against these downloaded plasmids using Geneious Prime version 2022.1.1, allowing us to generate the plasmid sequences referenced in our study. While we do not report similarity with the plasmids, we successfully recovered the plasmids as described in the manuscript.

For enhanced clarity, the plasmid sequences obtained through mapping have been submitted as supplementary files (S2 Data and S3 Data) in the current version of the manuscript. 

There's need for clarity on how many E. coli isolates were obtained from the 39 rats. How many isolates from each rat. Clarity is also required on the the 17 CREC. The MIC results of all 17 should be presented. The mcr-1 positive isolate was clearly colistin resistant (MIC of ≥ 8 µg/ml) but the others were susceptible ≤ 2 µg/mL according to EUCAST guidelines. Why were all 17 isolates considered CREC?

Response: We apologize for the inadequate and ambiguous description provided. In our study, we processed fecal samples by suspending 3 to 4 pellets in 1 ml of PBS and directly inoculated them onto MacConkey agar plates supplemented with colistin sulfate, without prior enrichment. It's worth noting that not all fecal samples yielded growth.

A total of 40 E. coli colonies were recovered from the 39 rats examined and were confirmed using E. coli-specific malB gene targeted PCR. Subsequently, all colonies underwent screening for mcr genes and antimicrobial resistance through disc diffusion. Interestingly, colonies originating from the same rat exhibited identical patterns on mcr PCR and antimicrobial resistance, leading us to consider them as a single entity. Following these criteria, we identified 17 colistin-resistant E. coli (CREC) isolates from 17 positive rats. 

Among these isolates, only those testing positive for mcr-1 displayed a minimum inhibitory concentration (MIC) of ≥ 8 µg/ml, while all others exhibited MIC values of ≥ 2 µg/ml (i.e., < 4 µg/ml). However, we did not determine the MIC fraction between 2 and 4 µg/ml. Considering the breakpoints established by the Clinical and Laboratory Standards Institute (CLSI) and the European Committee on Antimicrobial Susceptibility Testing (EUCAST), all isolates were deemed resistant to colistin, thereby classifying the 17 isolates as CREC.

Furthermore, it's important to highlight that according to CLSI guidelines, polymyxin resistance is categorized as intermediate or resistant, with no susceptible category. It's crucial to note that the intermediate category does not imply sensitivity but rather resistance to the antimicrobial agent.

Response to the Editorial comments:

Journal Requirements:

When submitting your revision, we need you to address these

additional requirements.

1. Please ensure that your manuscript meets PLOS ONE's style

requirements, including those for file naming. The PLOS ONE style templates can

be found at

and

Response: We have visited the link and prepared the manuscript accordingly.

2. Did you know that

depositing data in a repository is associated with up to a 25% citation

advantage (https://doi.org/10.1371/journal.pone.0230416)? If you’ve not already

done so, consider depositing your raw data in a repository to ensure your work

is read, appreciated and cited by the largest possible audience. You’ll also

earn an Accessible Data icon on your published paper if you deposit your data

in any participating repository (https://plos.org/open-science/open-data/#accessible-data).

Response: Thank you very much for the suggestion. We will think it over later. 

3. We suggest you thoroughly copyedit your manuscript for

language usage, spelling, and grammar. If you do not know anyone who can help

you do this, you may wish to consider employing a professional scientific

editing service.

Response: The manuscript was thoroughly copyedited for language usage, spelling, and grammar by one native speaker Dr. Andrew Crombie, School of Environmental Sciences, University of East Anglia, Norwich, NR4 7TJ, UK. A version of the copyedited manuscript will be provided for your kind perusal.

“This research was

conducted with partial support from the research grants provided to JH, by

Ministry of Education (MoE), Government of Bangladesh, Grant No. LS20191223;

and Bangladesh Agricultural University Research Systems (BAURES), grant No.

2021/21/BAU. The funders did not play any role in the study design, data

collection and analysis, decision to publish, or preparation of the manuscript.”

Please provide an amended statement that declares *all* the

funding or sources of support (whether external or internal to your

organization) received during this study, as detailed online in our guide for

authors at http://journals.plos.org/plosone/s/submit-now. Please also include the statement “There was

no additional external funding received for this study.” in your updated

Funding Statement.

Please include your amended Funding Statement within your

cover letter. We will change the online submission form on your behalf.

Response: The Funding statement is included in the Cover letter for your kind consideration.

5. Thank you for stating the following in the

Acknowledgments Section of your manuscript:

“The authors are indebted to the Ministry of Education (MoE),

Government of Bangladesh (Project No. LS20191223), and Bangladesh Agricultural

University Research Systems (BAURES) (Project No. 2021/21/BAU) for providing partial

fund for this research.”

We note that you

have provided additional information within the Acknowledgements Section that

is not currently declared in your Funding Statement. Please note that funding

information should not appear in the Acknowledgments section or other areas of

your manuscript. We will only publish funding information present in the

Funding Statement section of the online submission form.

Please remove any funding-related text from the manuscript

and let us know how you would like to update your Funding Statement. Currently,

your Funding Statement reads as follows:

“This research was conducted with partial support from the

research grants provided to JH, by Ministry of Education (MoE), Government of

Bangladesh, Grant No. LS20191223; and Bangladesh Agricultural University

Research Systems (BAURES), grant No. 2021/21/BAU. The funders did not play any

role in the study design, data collection and analysis, decision to publish, or

preparation of the manuscript.”

Please include your amended statements within your cover

letter; we will change the online submission form on your behalf.

Response: The Acknowledgement section was removed from the manuscript for your kind consideration.

6. We note that you have included the phrase “data not

shown” in your manuscript. Unfortunately, this does not meet our data sharing

requirements. PLOS does not permit references to inaccessible data. We require

that authors provide all relevant data within the paper, Supporting Information

files, or in an acceptable, public repository. Please add a citation to support

this phrase or upload the data that corresponds with these findings to a stable

repository (such as Figshare or Dryad) and provide and URLs, DOIs, or accession

numbers that may be used to access these data. Or, if the data are not a core

part of the research being presented in your study, we ask that you remove the

phrase that refers to these data.

Response: Thank you very much for your suggestion. As the data are not a core part of the research we have removed the phrase from the text. 

Additional Editor Comments (if provided):

Please carefully address the reviewers' comments from the two reviewers. Also, please cite the bioinformatics tools used in your study. For example, Line 129, BacWGSTdb 2.0 server.

Response: We are grateful for your kind suggestions. The reviewers' comments were addressed carefully. In addition, we have cited all the bioinformatics tools used in this study following your suggestions.

---

## [Decision Letter · Decision Letter 2]

16 Apr 2024

First detection and characterization of mcr-1 colistin resistant E. coli from wild rat in Bangladesh

PONE-D-23-39379R2

Dear Dr. Hassan,

We’re pleased to inform you that your manuscript has been judged scientifically suitable for publication and will be formally accepted for publication once it meets all outstanding technical requirements.

Kind regards,

Zhi Ruan, Ph.D.

Academic Editor

PLOS ONE

Additional Editor Comments (optional):

Reviewers' comments:

Reviewer's Responses to Questions

**Comments to the Author**

1. If the authors have adequately addressed your comments raised in a previous round of review and you feel that this manuscript is now acceptable for publication, you may indicate that here to bypass the “Comments to the Author” section, enter your conflict of interest statement in the “Confidential to Editor” section, and submit your "Accept" recommendation.

Reviewer #2: All comments have been addressed

Reviewer #3: All comments have been addressed

2. Is the manuscript technically sound, and do the data support the conclusions?

Reviewer #2: Yes

Reviewer #3: Yes

3. Has the statistical analysis been performed appropriately and rigorously? 

Reviewer #2: N/A

Reviewer #3: Yes

4. Have the authors made all data underlying the findings in their manuscript fully available?

Reviewer #2: No

Reviewer #3: Yes

5. Is the manuscript presented in an intelligible fashion and written in standard English?

Reviewer #2: Yes

Reviewer #3: Yes

6. Review Comments to the Author

Reviewer #2: (No Response)

Reviewer #3: (No Response)

7. PLOS authors have the option to publish the peer review history of their article (what does this mean?). If published, this will include your full peer review and any attached files.

Reviewer #2: No

Reviewer #3: **Yes: **Joshua Mbanga
